# A programmable DNA roadblock system using dCas9 and multivalent target sites

**Emily K. Matozel[1], Stephen Parziale[2], Allen C. Price[3]***

**1** Department of Biology, Emmanuel College, Boston, United States of America, **2** Department of Mathematics, Emmanuel College, Boston, United States of America, **3** Department of Chemistry and Physics, Emmanuel College, Boston, United States of America

\* priceal@emmanuel.edu

## Abstract

A protein roadblock forms when a protein binds DNA and hinders translocation of other DNA binding proteins. These roadblocks can have significant effects on gene expression and regulation as well as DNA binding. Experimental methods for studying the effects of such roadblocks often target endogenous sites or introduce non-variable specific sites into DNAs to create binding sites for artificially introduced protein roadblocks. In this work, we describe a method to create programmable roadblocks using dCas9, a cleavage deficient mutant of the CRISPR effector nuclease Cas9. The programmability allows us to custom design target sites in a synthetic gene intended for *in vitro* studies. These target sites can be coded with multivalency—in our case, internal restriction sites which can be used in validation studies to verify complete binding of the roadblock. We provide full protocols and sequences and demonstrate how to use the internal restriction sites to verify complete binding of the roadblock. We also provide example results of the effect of DNA roadblocks on the translocation of the restriction endonuclease NdeI, which searches for its cognate site using one dimensional diffusion along DNA.

**Data Availability Statement:** All relevant data are within the manuscript and its Supporting Information files or available for download from https://github.com/priceal/roadblocks-dCas9-data.

## 1. Introduction

Translocating DNA binding proteins (DBPs) face a landscape crowded with numerous other competing DBPs—some mobile, others static—which can interfere with one dimensional (1D) motion. For example, in euchromatin histone complexes separate strands of "naked" DNA, one to a few hundred base pairs in length, along which translocation is limited in the absence of hopping or other mechanism allowing for roadblock bypass. Other biologically relevant examples of DNA roadblocks include nucleic acid polymerases, transcription factors as well as repressor proteins. Such roadblocks can have significant biological consequences. For example, blocking DNA replication by Cascade is an important contributor to genomic instability of foreign DNA under attack by CRISPR [1]. In the case of DNA mismatch repair, specific mechanisms have evolved to allow repair proteins to bypass roadblocks [2].

In many investigations artificially introduced roadblocks have been used to probe protein-DNA interactions. In some cases, naturally occurring binding sites for DBPs have been used

**Funding:** ACP received grant # MCB-2120878 from the National Science Foundation (www.nsf.gov). The funders have not had and will not have a role in study design, data collection and analysis, decision to publish, or preparation of the manuscript.

**Competing interests:** The authors have declared that no competing interests exist.

for placing roadblocks. For example, the ability of RecBCD to displace roadblocks was investigated using endogenous binding sites in the λ phage genome for RNA polymerase, LacI, nucleosomes and a cleavage deficient mutant of the restriction endonuclease EcoRI [3]. Studies of translocation by the DNA repair enzyme MutL made use of RNA polymerase as well as dCas9 to target pre-existing sites in DNAs derived from phage DNA [4]. DNA replication has been shown to be blocked by dCas9 targeting naturally occurring sites both *in vitro* in substrates derived from plasmid DNA [5], and *in vivo* in yeast [6]. In other techniques artificially introduced binding sites have been used to strategically place roadblocks. These have included another study of the protein MutL, in which an EcoRI site was introduced into phage DNA using mutagenesis [7], and an *in vivo* study of DNA target search by LacI, in which TetR sites were engineered into the E. Coli genome to block 1D diffusion [8]. Other methods explored have employed the bacterial Ter/Tus replication fork barrier system [9] and covalent protein-DNA complexes [10].

The RNA guided nucleases, such as Cas9 and Cas12, have revolutionized genetic science by providing programmable "scissors" for modifying genetic information in a specific manner. Cleavage deficient mutants, such as dCas9, have allowed design of targeted DBPs either lacking or with alternate enzymatic function. Due to its high specificity and programmability, this system is a natural choice for a targeted artificial roadblock.

In this work, we show a method for creating a programmable dCas9 roadblock intended for use in interfering with translocation by DBPs *in vitro*. The method uses a synthetic gene which contains an artificial sequence targeted by dCas9 that is activated with a complementary sgRNA. We make use of the programmability of dCas9 to encode a multivalent roadblock site. In this case, the target sequence is designed with internal restriction sites to allow for direct validation of complete binding by the roadblock—a crucial requirement for effectiveness. We provide complete protocols and sequences, as well as sample validation and experimental data. For the sample data, we use a single molecule DNA tethering method we have previously used to measure the Brownian motion of DNA [11], DNA replication [12], as well as DNA target search [13] that provides single molecule resolution.

## 2. Materials and methods

The protocol for producing roadblocked DNA substrates described in this peer-reviewed article, including all sequences, is published on protocols.io (https://www.protocols.io/view/a-programmable-dna-roadblock-system-using-dcas9-an-36wgq422kvk5/v2) and is included for printing as S1 File with this article. All data presented in this work can be downloaded from https://github.com/priceal/roadblocks-dCas9-data.

Roadblocks were tested using a single molecule kinetic assay for site specific DNA cleavage described in detail elsewhere [14, 15]. Here we briefly describe the main steps. In the PCR described in the protocol, all forward primers were labeled with digoxigenin and reverse primers with biotin. Either roadblocked or naked DNA (without roadblocks) was used to tether 1 μm magnetic beads in a microfluidic channel (Fig 2A). The DNA-tethered beads were observed under low magnification and dark field imaging. Typically, several hundred beads were visible in a single experiment. A magnet was used to apply a small force (~50 fN) to the beads, which were pulled out of focus as the DNA was cleaved. Samples containing enzyme were injected in reaction buffer (20 mM Tris-HCl, 20–100 mM NaCl, 1 mg/mL β-casein, 1 mg/mL Pluronic F-127, 2 mM MgCl$_2$). The number of beads in each image was counted to determine fractions of uncleaved DNA and quantitative analysis was done as described elsewhere [16].

Gel based assays were utilized to confirm the specificity of prepared sgRNA. Cas9 was activated with the sgRNA and reacted with DNA in the same method as described in the protocol for dCas9, with the exception that Cas9 was used in the reaction mix instead of dCas9. After the cleavage reaction and prior to the addition of loading dye, the mixture was treated with Proteinase K for 10 minutes at 25˚C in order to degrade any remaining Cas9 bound to the cleaved DNA.

## 3. Expected results

### 3.1. Specificity of roadblock

The specificity of the dCas9 roadblock is determined by the sgRNA sequence. We verified this specificity by activating cleavage competent Cas9 with our sgRNAs and incubating the activated Cas9 with our substrate DNA. If the sgRNA is specific for our roadblock sites, then the substrate will be cleaved at positions 400 and 600 in the 1000 bp DNA, resulting in fragments of length 400 and 200 bp. Experimental results, shown in Fig 1 confirm this pattern.

### 3.2. Complete binding of roadblock

In order to be effective, the roadblock must bind 100% of targeted sites among all substrate DNA molecules. This can be verified by making use of the BtgI site which is encoded in the sequence targeted by the sgRNA. Upon complete binding of the roadblock, no cleavage by BtgI should be possible. We verified this with a parallel high throughput single molecule

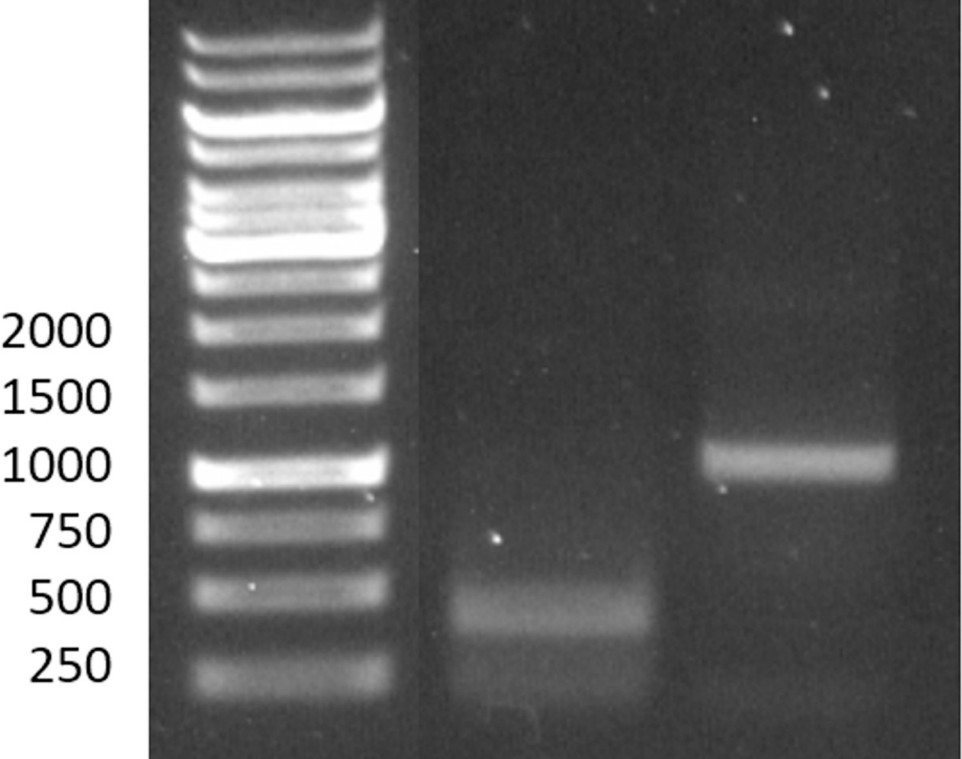

**Fig 1. Specificity of roadblock.** Lane 1 is the DNA ladder (size in base pairs indicated at left). Lane 2 shows the result of cleavage of DNA with activated Cas9. Lane 3 shows DNA without Cas9 present. Note, section of image showing irrelevant lanes between ladder and Cas9 sample (lane 2) has been removed (see S1 Fig for complete gel image).

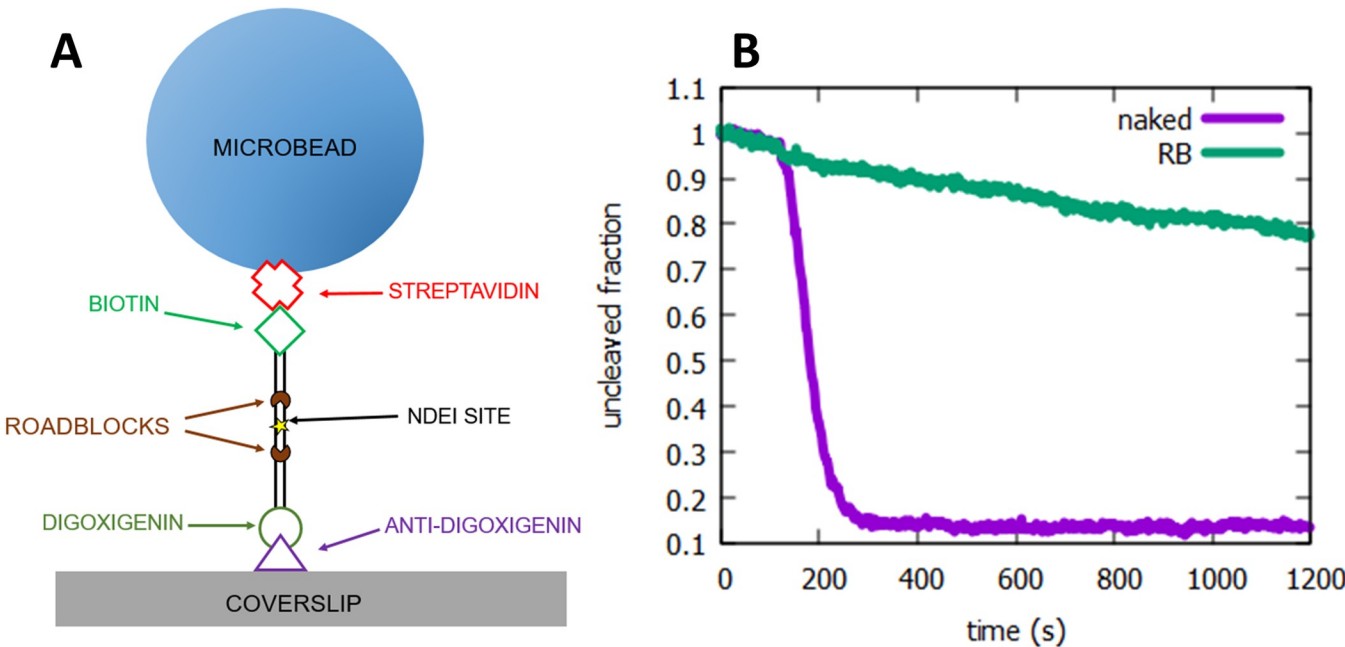

**Fig 2. Complete binding of roadblock.** (A) Schematic showing DNA tethers with the locations of NdeI cognate and roadblock sites indicated. BtgI sites are contained within the roadblock sites. (B) Plot of fraction of uncleaved DNA versus time after injection of BtgI is shown. Purple curve is naked DNA without roadblocks. Rapid decrease at 180s is due to cleavage of the internal BtgI site contained in the roadblock sequence. Green curve is DNA with roadblocks. No cleavage is observed after injection of BtgI (180s). The gradual decrease over time is due to DNA tethered beads detaching from the surface non-specifically, and not due to DNA cleavage.

kinetic assay [15]. Substrate DNAs, measuring 1000 bp and containing two roadblock sites at positions 400 bp from either terminus, were used to tether microbeads (diameter 1μm) in a microfluidic cell (Fig 2A). Injection of BtgI resulted in immediate cleavage of naked DNA, while no cleavage was observed when DNA bound with roadblocks was used (Fig 2B). Any incomplete binding by roadblocks would lead to a rapid drop in bead count upon injection of BtgI (at roughly 180s) in the roadblocked DNA, which was not observed.

### 3.3. Blocking DNA translocation

In order to perform the intended function, a roadblock must interfere with protein translocation along DNA. We demonstrated this by observing the kinetics of cleavage by NdeI under diffusion controlled conditions ([NdeI] = 20 pM) in the presence and absence of roadblocks. The DNA target search strategy of NdeI, like that of many restriction endonucleases, relies on one dimensional diffusion along DNA [13, 17]. Therefore, effective roadblocks are predicted to slow the target search kinetics when the diffusion length of the protein along the DNA is at least as great as the distance from the NdeI cognate site to the roadblock. We demonstrated this slowing by measuring the DNA cleavage kinetics of a centrally located NdeI recognition site in the presence and absence of roadblocks on either side of the NdeI site. Roadblocks significantly slow target search in 20 mM NaCl (Fig 3A) where the target search is known to depend strongly on 1D diffusion. However, in 80 mM NaCl (Fig 3B) where the target search is controlled by 3D diffusion, there is little effect. This increase in search time is due to the blockage of paths that would lead to the cognate site in the absence of the roadblocks, hence demonstrating the utility of dCas9 to block 1D translocation along DNA.

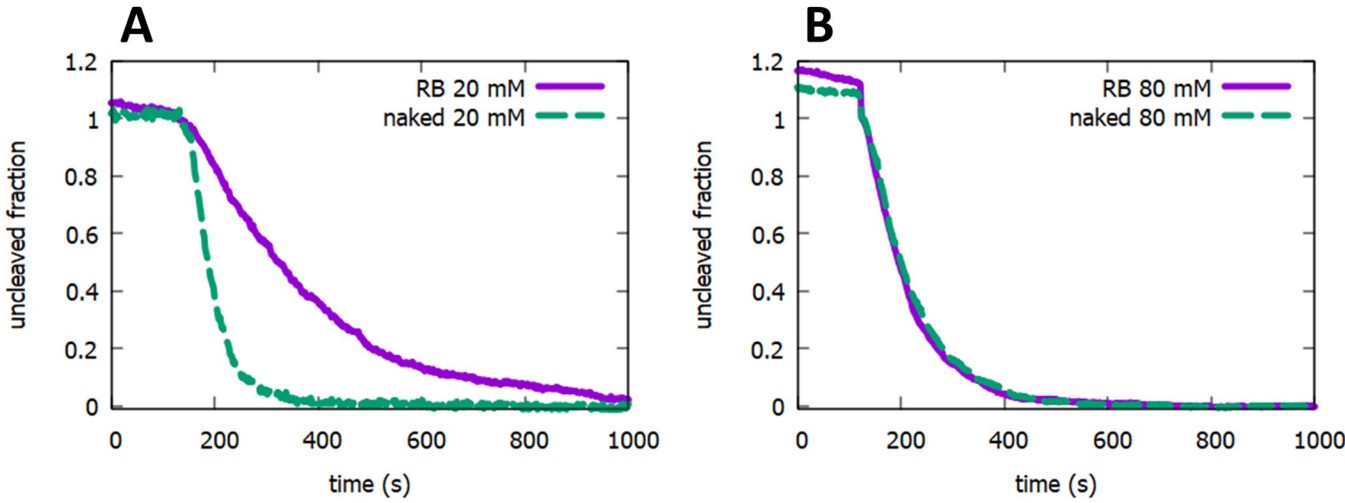

**Fig 3. Effect of roadblocks on DNA translocation.** Cleavage by NdeI under target search limiting conditions for both naked (dashed green curve) and roadblocked (solid purple) DNA is shown. (A) 20 mM NaCl. Target search relies on extensive translocation events and roadblocks have significant effect. (B) 80 mM NaCl. Target search makes use of short translocation events and roadblocks have no effect. Data is scaled in each case so that the uncleaved fraction is one at time of NdeI injection (180s).

### 3.4. Non-specific binding by apo-dCas9 is not observed

Several studies have indicated that unactivated (apo-Cas9), that is, Cas9 that is not bound to sgRNA, can bind non-specifically to DNA. Non-specific binding could potentially lead to non-specific "roadblocking" of translocation. We tested for the presence of this non-specific binding by first incubating our DNAs with apo-dCas9 extensively before tethering. After this incubation, tethering and injection of NdeI were carried out as described above. The results for NdeI cleavage after apo-dCas9 incubation, compared to control experiments in the absence of apo-dCas9 are shown in Fig 4. Both low salt (40 mM) where significant translocation is known to occur, and high salt (100 mM) conditions were tested. The lack of difference between the

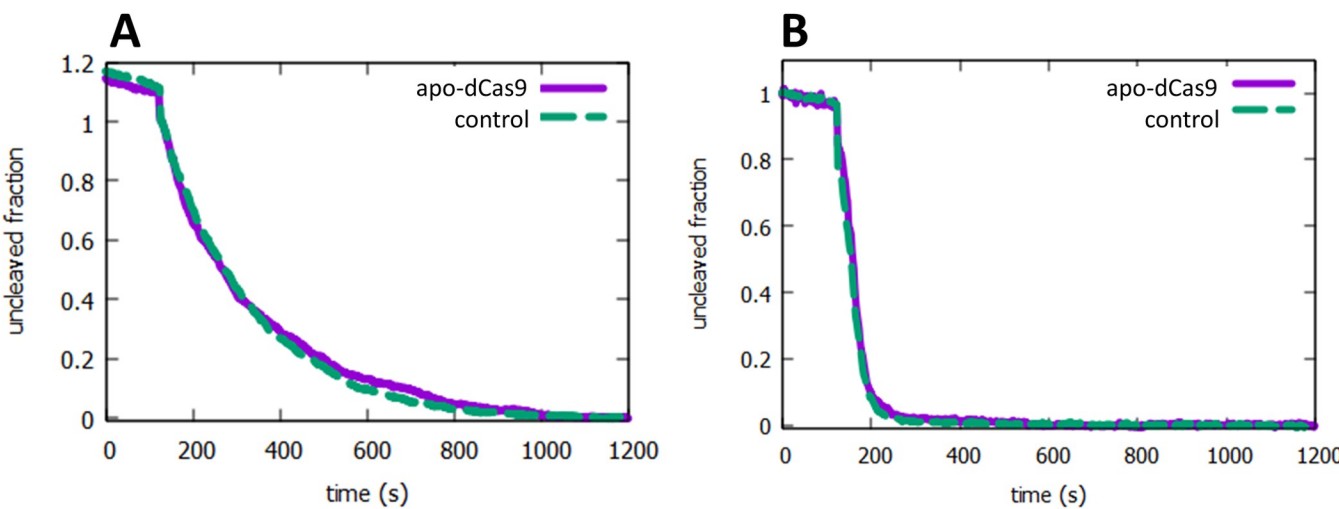

**Fig 4. Non-specific binding by apo-dCas9 is not observed.** Cleavage by NdeI under target search limiting conditions with both naked (control, dashed green curve) and DNA incubated with apo-dCas9 (apo-dCas9, purple solid curve) is shown. (A) 40 mM NaCl. (B) 100 mM NaCl.

results with the naked and roadblocked DNA supports the conclusion that non-specific binding of apo-dCas9 is not significant in these experiments.

## 4. Discussion

Although the method described here is intended for use *in vitro*, it can be extended to *in vivo* applications with appropriate modifications. For example, gene editing could be used to introduce the custom target site into a genome. Specificity of the roadblocks could be verified using Chromatin Immunoprecipitation (ChIP).

## Supporting information

**S1 File.**
(PDF)

**S1 Fig.**
(PDF)

## Acknowledgments

The authors would like to thank Emma E. Stevens, Van T. Nguyen and Jaqueline R. Ferreira for useful discussions on manuscript.

## Author Contributions

**Conceptualization:** Allen C. Price.

**Funding acquisition:** Allen C. Price.

**Investigation:** Emily K. Matozel, Stephen Parziale, Allen C. Price.

**Writing – original draft:** Emily K. Matozel, Stephen Parziale, Allen C. Price.

**Writing – review & editing:** Emily K. Matozel, Stephen Parziale, Allen C. Price.

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
