## [Decision Letter · Decision Letter 0]

18 Mar 2022

PONE-D-22-04792A programmable DNA roadblock system using dCas9 and multivalent target sitesPLOS ONE

Dear Dr. Price,

Thank you for submitting your manuscript to PLOS ONE. After careful consideration, we feel that it has merit but does not fully meet PLOS ONE’s publication criteria as it currently stands. Therefore, we invite you to submit a revised version of the manuscript that addresses the points raised during the review process.

The criticisms of the two reviewers are relatively minor and can be easily addressed. I too reviewed this manuscript and do not have additional points of criticism  beyond those of the reviewers. Including the recommended revisions in the revised manuscript will improve it.  The addition of schematic presentations of the protocols would greatly enhance a readers understanding of the details of the protocols. Please respond to all of the criticisms in your response to the editors, either indicating how you have addressed them or, if you do not agree to the revisions, a rebuttal to the reviewers comments.

We look forward to receiving your revised manuscript.

Kind regards,

Michael R Volkert, Ph.D.

Academic Editor

PLOS ONE

Journal Requirements:

"This work was supported by National Science Foundation grant MCB-2120878. The authors would like to thank Emma E. Stevens, Van T. Nguyen and Jaqueline R. Ferreira for useful discussions on manuscript."

"ACP received grant # MCB-2120878 from the National Science Foundation (www.nsf.gov). The funders have not had and will not have a role in study design, data collection and analysis, decision to publish, or preparation of the manuscript."

Reviewers' comments:

Reviewer's Responses to Questions

**Comments to the Author**

1. Does the manuscript report a protocol which is of utility to the research community and adds value to the published literature?

Reviewer #1: Yes

Reviewer #2: Yes

2. Has the protocol been described in sufficient detail?

Descriptions of methods and reagents contained in the step-by-step protocol should be reported in sufficient detail for another researcher to reproduce all experiments and analyses. The protocol should describe the appropriate controls, sample sizes and replication needed to ensure that the data are robust and reproducible.

Reviewer #1: Partly

Reviewer #2: Yes

3. Does the protocol describe a validated method?

Reviewer #1: Yes

Reviewer #2: Yes

4. If the manuscript contains new data, have the authors made this data fully available?

Reviewer #1: Yes

Reviewer #2: Yes

**5. Is the article presented in an intelligible fashion and written in standard English?**

Reviewer #1: Yes

Reviewer #2: Yes

6. Review Comments to the Author

Reviewer #1: In this manuscript, Matozel et al describe a protocol to create in vitro roadblocks using deadcas9. This is an important area to study DNA protein barrier-related DNA metabolism including DNA repair and transcription. I have a few comments:

(1) In the Introduction, please cite novel work from Scully lab where they used bacteria system Tus/Ter to create DNA- protein block in a mammalian system and studied how DNA translocase protein FANCM regulate repair pathway choice at DNA protein barrier (Panday et al, Molecular Cell 2021). Please also cite the work from Natalia Y Tretyakova lab on using 5- Formylcytosine mediated DNA- Protein block Shaofei Ji et al Nucleic Acids research 2018. Also Goro Doi et al Nucleic acids research 2021 where they used dcas9 to impair fork progression.

(2) Optional: Author Could use Chromatin Immunoprecipitation (CHIP) to study the specificity of roadblocks. If authors cannot do that experiment, please mention this in the discussion section as another way to confirm specificity.

(3) Please mention the Sg sequence and highlight PAM sequence.

Reviewer #2: In this work, Matozel et al describe a detailed protocol to investigate programmable binding of dCas9 on double stranded DNA substrates and consequent blocking of diffusing/translocating molecules occupying the DNA. Understanding how multiple proteins simultaneously navigate/process DNA is an important question, and the use of a programmable roadblock as a model system lends itself to future investigation. The protocols are detailed and the accompanying experiments are robust, and their interpretation is sensible. The manuscript is well written and is very easy to follow. Overall, this is work that would benefit others in the field to answer mechanistic questions in protein-DNA interactions.

I only have one major comment which is that all figures should be accompanied by 'schematics' of the experimental setup (even if they have been described elsewhere). Without these schematics, naive readers may not be able to sufficiently visualize the experiment, and contextualize the results.

7. PLOS authors have the option to publish the peer review history of their article (what does this mean?). If published, this will include your full peer review and any attached files.

Reviewer #1: No

Reviewer #2: No

---

## [Author Response · Author response to Decision Letter 0]

15 Apr 2022

Please see attached Response to Reviewers document.

I have deposited all data in a publicly accessible database as requested. I have added an additional line to the first paragraph of the materials and methods sections that reads "All data presented in this work can be downloaded from https://github.com/priceal/roadblocks-dCas9-data ."

I have also added labels A and B to figure 3 panels.

---

## [Editor Report · Decision Letter 1]

22 Apr 2022

A programmable DNA roadblock system using dCas9 and multivalent target sites

PONE-D-22-04792R1

Dear Dr. Price,

We’re pleased to inform you that your manuscript has been judged scientifically suitable for publication and will be formally accepted for publication once it meets all outstanding technical requirements.

Kind regards,

Michael R Volkert, Ph.D.

Academic Editor

PLOS ONE
---

## [Editor Report · Acceptance letter]

27 Apr 2022

PONE-D-22-04792R1 

A programmable DNA roadblock system using dCas9 and multivalent target sites 

Dear Dr. Price:

I'm pleased to inform you that your manuscript has been deemed suitable for publication in PLOS ONE. Congratulations! Your manuscript is now with our production department. 

Kind regards, 

on behalf of

Prof. Michael R Volkert 

Academic Editor

PLOS ONE